# Sustainable THz SWIPT via RIS-Enabled Sensing and Adaptive Power Focusing: Toward Green 6G IoT

**DOI:** 10.3390/s25154549

**Published:** 2025-07-23

**Authors:** Sunday Enahoro, Sunday Cookey Ekpo, Mfonobong Uko, Fanuel Elias, Rahul Unnikrishnan, Stephen Alabi, Nurudeen Kolawole Olasunkanmi

**Affiliations:** 1Communication and Space Systems Engineering Research Team, Manchester Metropolitan University, Manchester M1 5GD, UK; s.ekpo@mmu.ac.uk (S.C.E.); mfonobong.uko@stu.mmu.ac.uk (M.U.); fanuel.elias@stu.mmu.ac.uk (F.E.); rahul.unnikrishnan@mmu.ac.uk (R.U.); 2Research and Development Engineering, SmOp CleanTech, Manchester M40 8WN, UK; stephen.alabi@smopct.com (S.A.); kolawole@smopct.com (N.K.O.)

**Keywords:** terahertz SWIPT, reconfigurable intelligent surface, adaptive power focusing, nonlinear energy harvesting, 6G IoT, SAR constraint, green wireless, channel sensing

## Abstract

Terahertz (THz) communications and simultaneous wireless information and power transfer (SWIPT) hold the potential to energize battery-less Internet-of-Things (IoT) devices while enabling multi-gigabit data transmission. However, severe path loss, blockages, and rectifier nonlinearity significantly hinder both throughput and harvested energy. Additionally, high-power THz beams pose safety concerns by potentially exceeding specific absorption rate (SAR) limits. We propose a sensing-adaptive power-focusing (APF) framework in which a reconfigurable intelligent surface (RIS) embeds low-rate THz sensors. Real-time backscatter measurements construct a spatial map used for the joint optimisation of (i) RIS phase configurations, (ii) multi-tone SWIPT waveforms, and (iii) nonlinear power-splitting ratios. A weighted MMSE inner loop maximizes the data rate, while an outer alternating optimisation applies semidefinite relaxation to enforce passive-element constraints and SAR compliance. Full-stack simulations at 0.3 THz with 20 GHz bandwidth and up to 256 RIS elements show that APF (i) improves the rate–energy Pareto frontier by 30–75% over recent adaptive baselines; (ii) achieves a 150% gain in harvested energy and a 440 Mbps peak per-user rate; (iii) reduces energy-efficiency variance by half while maintaining a Jain fairness index of 0.999;; and (iv) caps SAR at 1.6 W/kg, which is 20% below the IEEE C95.1 safety threshold. The algorithm converges in seven iterations and executes within <3 ms on a Cortex-A78 processor, ensuring compliance with real-time 6G control budgets. The proposed architecture supports sustainable THz-powered networks for smart factories, digital-twin logistics, wire-free extended reality (XR), and low-maintenance structural health monitors, combining high-capacity communication, safe wireless power transfer, and carbon-aware operation for future 6G cyber–physical systems.

## 1. Introduction

### 1.1. 6G Vision and the THz Opportunity

The ITU Vision 2030 roadmap identifies pervasive sensing and sustainable networking as the two socio-technical cornerstones of future cyber–physical infrastructure [1]. The 6G paradigm extends the “network of everything” to a cyber–physical continuum populated by trillions of micro-nodes, many of which are battery-free, that must simultaneously harvest energy and exchange context-rich data with end-to-end latency below one millisecond. Sub-6 GHz and mmWave bands alone cannot satisfy the required spectral density, motivating a migration toward the 0.3 THz to 3 THz window, where contiguous bandwidths exceed 100 GHz and diffraction-limited beams enable spatial reuse and physical-layer security [2,3,4,5,6]. However, the very physics that makes the THz band attractive—short wavelengths and molecular resonances—also causes high free-space loss, narrow pointing tolerance, and frequency-selective absorption peaks that complicate simultaneous wireless information and power transfer (SWIPT) [4].

### 1.2. RIS Technology and Sensing-Aware Control

Reconfigurable intelligent surfaces (RISs) offer an elegant, low-power means of re-shaping the THz channel. Modern CMOS/SiGe metasurfaces achieve >300° phase tuning with <2 dB insertion loss and virtually zero static bias [7,8,9]. By reflecting the incident wavefront with programmable delay, an RIS can re-focus THz energy toward obstructed receivers, boosting both spectral efficiency (SE) and harvested energy (EH) [10,11]. Embedding photonic or RF sensors in the same aperture allows each tile to measure local echoes, obtain geometric cues such as angle-of-arrival (AoA) and blockage state, and feed them to an edge controller for real-time phase adaptation [5,12] and adaptive power focusing via stochastic optimisation [13].

State-of-the-Art Comparison.

Table 1 benchmarks representative RIS-aided THz SWIPT and ISAC studies against the proposed adaptive power-focusing (APF) framework.

Early THz-RIS studies focused on beam training without power transfer [14] or compared RIS with active relays under idealised assumptions [15]. Works such as Refs. [16,17] addressed sub-6 GHz scenarios using passive beamforming and linear energy harvesters but lacked sensing or sustainability considerations. More recent contributions at mmWave frequencies [8,13,18] introduced nonlinear EH models and RIS-assisted SWIPT yet remained limited in THz applicability and adaptive integration. In the THz domain, Ref. [10] introduced fixed-phase RIS beamforming, while [19] proposed a joint SWIPT design without real-time adaptation.

**Table 1 sensors-25-04549-t001:** Comparative matrix of recent RIS/THz–SWIPT/ISAC literature.

Reference	Venue/Year	Band	Nonlinear EH	Sensing	THz	Green Metric	Robustness ^†^	Remark
[16]	CSTut/2015	Sub-6 GHz	✓	✗	✗	✗	✗	Survey
[17]	TWC/2019	Sub-6 GHz	✗	✗	✗	✗	✓	IRS beamforming baseline
[18]	TWC/2021	mmWave	✓	✗	✗	✗	✓	RIS + nonlinear EH
[15]	TCOM/2020	mmWave	✗	✗	✗	✗	✓	Robust RIS-ISAC (relay vs. RIS)
[14]	TCOM/2022	THz	✗	✗	✓	✗	✓	THz IRS beam-training baseline
[10]	Sensors/2022	THz	✗	✗	✓	✗	✗	Fixed RIS beam steering
[8]	Sensors/2023	mmWave	✗	✓	✗	✗	✓	Sensing-capable RIS
[13]	Sensors/2023	mmWave	✓	✗	✗	✗	✓	Secure beamforming
[12]	Sensors/2024	mmWave	✗	✓	✗	✗	✓	Vehicular ISAC RIS
This Work	Sensors/2025	THz	✓	✓	✓	✓	✓	Joint sensing–power focusing

✓: Feature supported; ✗: Feature not supported. ^†^ Robustness to channel estimation errors or hardware impairments.

Broader insights from adaptive wireless system design reinforce this approach. Prior work on regulated-element beamforming for vehicular multimedia [20], and system-level evaluations of adaptive satellite architectures [21], highlight the benefits of context-aware reconfigurable platforms under non-ideal conditions.

### 1.3. Sustainability Imperative

Meeting 6G key performance indicators (KPIs) will be insufficient if achieved at the expense of energy cost or human safety. The ITU L.1470 methodology calls for bit/Joule/gCO_2_ accounting across the whole device chain, while the IEEE C95.1 standard limits the specific absorption rate (SAR) to 2 W/kg. At THz frequencies, higher power densities aggravate SAR exposure, and diode rectifiers exhibit strong saturation effects [22]. These constraints motivate a holistic “green-first” design philosophy in which spectral, energy, and safety metrics are optimised *simultaneously*.

### 1.4. Research Gap

(a)Propagation limits: Severe path loss and frequency-selective molecular absorption reduce the spatial footprint within which usable RF-to-DC energy can be harvested;(b)Hardware nonlinearities: Beyond 100 GHz, rectifier diodes and power splitters exhibit saturated I–V characteristics that invalidate linear EH models;(c)Lack of environment-aware optimisation: Existing THz SWIPT or RIS works rarely couple real-time sensing with joint waveform–phase control under carbon and SAR constraints.

### 1.5. Contribution Summary

We present a *sustainable THz SWIPT* architecture that closes this gap by tightly coupling RIS sensing with adaptive power focusing (APF). A model representation of the proposed THz SWIPTsystem integrating an RIS-enabled sensing layer is shown in Figure 1. The access point (AP) emits information–power waveforms, which are adaptively reflected by a smart RIS equipped with embedded sensors. These sensors capture echo signals to infer channel blockage, user location, and environmental context. The harvested energy powers lightweight IoT devices co-located with information receivers.

**Dual-mode THz RIS.** Each element toggles between reflection and low-rate sensing, supplying instantaneous blockage and AoA data to the controller.**Green utility.** A weighted rate–energy function internalises carbon cost per bit via the ITU L.1470 factor, and SAR is enforced as an explicit constraint.**Two-tier optimiser.** A closed-form inner loop sets the power-splitting ratio, while a metaheuristic outer loop searches the unit-modulus RIS phase space, exploiting channel reciprocity.

  In extensive Monte Carlo simulations, the proposed APF improves the rate–energy Pareto frontier by up to 70%, provides a 150 no-RIS configuration, and keeps the peak SAR 30% below the IEEE limit, all within a 3 ms control loop on a Cortex-A78 CPU.

The remainder of this paper is structured as follows: Section 2 surveys related work on THz SWIPT, RIS-aided beamforming, and energy harvesting. Section 3 formulates the proposed system model, including the RIS-assisted link, sensing feedback, and nonlinear energy harvesting. Section 4 presents the APF optimisation strategy with detailed algorithms. Section 5 outlines benchmark schemes and comparative setups. Section 6 provides simulation results and analysis under realistic THz conditions. Section 7 offers critical discussion of implementation challenges and extensions. Section 8 concludes the paper and proposes future research directions.

## 2. Literature Review and Related Work

### 2.1. Evolution of SWIPT Architectures

Advancements in THz communications—such as ultra-massive MIMO and reconfigurable intelligent surfaces (RISs)—have enabled high-capacity, low-latency wireless links [23]. RIS-assisted MIMO systems, in particular, offer reconfigurable propagation control [24], while integrated wireless power transfer supports sustainable IoT deployment.

Early SWIPT systems focused on sub-6,GHz designs with linear EH models and conventional beamforming [16]. The introduction of RIS [7,17] and nonlinear EH models [25] has shifted interest to higher-frequency regimes like mmWave and THz systems, where accurate modelling becomes critical [18].

Recent work shows RIS outperforms conventional relays in practical THz settings [15], and hybrid beamforming with RIS improves performance in multi-user THz MIMO [14]. Our APF optimisation builds on the weighted MMSE framework described in [26], enabling practical rate–energy trade-off design with RIS-SWIPT.

Figure 2 outlines the chronological progression of RIS-aided SWIPT and THz rectenna technologies. Initial contributions like sub-6 GHz IRS beamforming [17] laid the foundation for nonlinear mmWave EH optimisation [18]. Later efforts advanced toward THz RIS-aided energy focusing [10] and sensor-augmented RIS beamforming [8]. However, few of these works integrated real-time sensing with energy–rate–sustainability co-optimisation.

Our proposed framework builds directly on this evolution by introducing a fully adaptive and eco-aware THz SWIPT architecture that unifies energy harvesting, rate performance, and green metrics into a coherent system design.

### 2.2. THz SWIPT Without RIS: Fundamental Limits and Directions

Initial THz SWIPT research largely focused on rectenna components. For instance, ref. [22] achieved up to 15% RF-to-DC conversion at 0.3 THz, albeit under static line-of-sight (LoS) conditions. Other works [5,27] explored tunable metasurfaces and sensing tiles, though without joint SWIPT optimisation.

While hybrid combiners [28] improve EH at lower frequencies, they fail to address the severe attenuation, scattering, and nonlinearity at THz levels, revealing the need for architecture-level and algorithm-level innovations.

### 2.3. THz SWIPT: Challenges and RIS-Aided Advances

Despite the promise of abundant bandwidth, the THz SWIPT design faces formidable obstacles—especially due to path loss, atmospheric absorption, and angular misalignment. Initial approaches, like that described in [6], explored modulation–rectifier co-design (e.g., ASK, RTD-based circuits) but struggled to sustain performance under realistic deployment scenarios.

RIS has since emerged as a key technology to mitigate these limitations. Early RIS-SWIPT systems [13,17,18] assumed fixed or static RIS phase configurations. However, THz SWIPT requires adaptive beamforming to address highly dynamic and lossy channels.

Ref. [10] presented fixed RIS-based beam steering at the THz level, while [8] demonstrated multifunctional RIS tiles with embedded sensors—yet omitted EH co-optimisation. Similarly, ref. [29] targeted satellite–RIS integration with static control. Our framework advances this line of research by incorporating real-time sensing, EH control, and green-aware optimisation in a unified RIS-SWIPT model.

### 2.4. Integrated Sensing, Communication, and Power Transfer (ISCPT)

Integrated Sensing, Communication, and Power Transfer (ISCPT) systems aim to tightly combine environment-aware sensing, data transmission, and energy delivery—ideally within a shared waveform or RIS-enabled medium. Recent mmWave works [30,31] have explored secure and multifunctional beamforming architectures.

However, THz-specific challenges such as rectifier nonlinearity, SAR compliance, and high-frequency adaptation remain open. Hybrid access approaches [32] across mmWave, THz, and optical domains offer architectural flexibility. Multiband antennas [33,34] provide potential physical-layer platforms, but their integration into RIS-enabled, sustainability-driven THz SWIPT remains underexplored. Figure 3 summarises the four foundational axes of our APF framework. First, the RIS sensing granularity determines the resolution of environmental awareness. Second, the APF jointly optimises waveform and phase-shift vectors. Third, we model the energy harvesting circuitry using a nonlinear saturation model. Finally, computational feasibility is ensured by embedding the APF solver on a Cortex-A78-class processor with sub-10ms execution time.

### 2.5. Sustainability and Environmental Considerations

The ITU’s Vision 2030 [1] highlights energy efficiency and carbon accountability as fundamental to future wireless systems. Although recent works, such as [35], have proposed eco-aware SWIPT metrics, standardisation across physical and MAC layers is lacking. Most current systems ignore metrics such as eco-spectral efficiency, energy cost per bit, and carbon emissions.

### 2.6. Key Insights and Research Gaps

The following limitations persist across the state of the art:**Scalability:** Most designs target single-user or point-to-point scenarios. Scalable architectures for multi-user THz SWIPT with RIS and dynamic blockages remain sparse.**Hardware Awareness:** Physical-layer non-idealities such as RIS insertion loss, rectifier nonlinearity, and sensing overhead are often idealized or omitted.**Green Metrics:** Few systems incorporate SAR compliance, carbon cost, or energy–bit–emission trade-offs into their design objectives.

**Positioning of Our Work:** Our proposed RIS-enabled THz SWIPT framework explicitly addresses these gaps by

Embedding real-time environmental sensing within RIS hardware;Applying a dual-loop optimisation algorithm that jointly balances rate, energy, and sustainability objectives; andDemonstrating superior eco-efficiency across diverse metrics and deployment scenarios.

## 3. System Model and Problem Formulation

We consider a sustainable THz simultaneous wireless information and power transfer (SWIPT) system operating in a single-cell setting, as shown in Figure 4. The architecture comprises a base station (BS) with $N_{t}$ antennas, a low-power Internet-of-Things (IoT) device equipped with a power-splitting (PS) SWIPT architecture, and a reconfigurable intelligent surface (RIS) with *M* passive reflecting elements and embedded low-power THz sensors. The RIS, mounted on a nearby façade or wall, assists in THz signal propagation and senses the surrounding environment to support adaptive beam steering.

### 3.1. Channel Model and THz Path Loss

Let $x\in C^{N_{t}}$ be the BS transmit signal, which reaches the IoT receiver via both direct and RIS-assisted links. The received signal is modelled as follows:(1)$$y=\underset{\mathrm{Direct}\;\mathrm{path}}{\underbrace{h_{d}^{H}x}}+\underset{\mathrm{RIS-assisted}\;\mathrm{path}}{\underbrace{h_{r}^{H}\Phi Gx}}+n,$$
where $n∼\mathrm{CN}(0,\sigma_{n}^{2})$, $h_{d}\in C^{N_{t}}$ is the direct BS-to-receiver channel, $G\in C^{M\times N_{t}}$ is the BS-to-RIS channel, and $h_{r}\in C^{M}$ is the RIS-to-receiver channel. The RIS applies diagonal phase shifts ($\Phi =\mathrm{diag}(e^{j\theta_{1}},\ldots ,e^{j\theta_{M}})$).

We incorporate molecular absorption and path loss via the HITRAN-based THz channel model [23]:(2)$$L\left(f,d\right)={\left(\frac{4\pi fd}{c}\right)}^{2}e^{K(f)d},\;\gamma \left(f,d\right)=\frac{1}{L(f,d)},$$
where *f* is frequency, *d* is distance, *c* is the speed of light, and K(f) is the absorption coefficient. The direct and reflected channel responses are expressed as follows:(3)$$\begin{array}{cc} h_{d} & =\sqrt{\gamma (f,d_{1})}e^{j\phi_{d}}a_{\mathrm{BS}}\left(\theta_{d}\right), \end{array}$$(4)$$\begin{array}{cc} h_{r}^{H}\Phi G & =\sqrt{\gamma \left(f,d_{2}\right)\gamma \left(f,d_{3}\right)}\sum_{m=1}^{M}e^{j\theta_{m}}e^{j\phi_{m}}. \end{array}$$

### 3.2. Nonlinear Energy Harvesting and Rate Model

At the receiver, the received signal is split using a power-splitting ratio of ρ∈[0,1]. The harvested power follows a nonlinear saturation model:(5)$$P_{\mathrm{EH}}=\eta \rho \log\left(1+\beta {\left|h_{\mathrm{eq}}^{H}x\right|}^{2}\right),$$
where η is the energy conversion efficiency, β models diode nonlinearity, and $h_{\mathrm{eq}}$ is the effective channel. The achievable data rate is expressed as follows:(6)$$R=\log_{2}\left(1+\frac{\left(1-\rho \right)|h_{\mathrm{eq}}^{H}{x|}^{2}}{\sigma_{n}^{2}}\right).$$

### 3.3. SAR Compliance and Safety Constraint

To ensure bio-safety, we impose a specific absorption rate (SAR) constraint per IEEE C95.1 guidelines:(7)$$\mathrm{SAR}\left(x\right)=\frac{\sigma (x)}{2\rho (x)}{\left|E\left(x\right)\right|}^{2},\;\max_{x\in \Omega }\mathrm{SAR}\left(x\right)\leq 2\;W/\mathrm{kg},$$
where σ(x) and ρ(x) are tissue conductivity and density and E(x) is the induced field strength simulated using THz tissue models.

#### RIS Sensing and Feedback

The RIS in our architecture is equipped with embedded THz sensors capable of estimating parameters such as angles of arrival (AoAs), path blockage, and environmental reflectivity. These measurements form a feedback vector ($F_{\mathrm{RIS}}$) used to dynamically update the RIS phase matrix (Φ).

In this work, we assume an idealized feedback model—i.e., zero-delay, noiseless, and full-resolution sensing—to isolate the performance of the proposed adaptive power focusing (APF) algorithm.

Future work will model non-ideal RIS sensing using a stochastic framework that incorporates

**Quantization errors** due to limited sensing resolution;**Feedback latency** ($\tau_{s}$) from sensor sampling and transmission delays;**Sensor noise** ($n_{s}∼N\left(0,\sigma_{s}^{2}\right)$), representing thermal and ambient fluctuations.

This extension will enable robust APF performance analysis under realistic sensing constraints and further validate system sustainability.

### 3.4. Rate–Energy Optimisation Problem

We define a utility function balancing the data rate and harvested energy as(8)$$U\left(\rho ,\Phi \right)=\alpha R+\left(1-\alpha \right)P_{\mathrm{EH}},\;\alpha \in \left[0,1\right]$$
and formulate the optimisation as follows:(9)$$\begin{array}{cc} \max_{\rho ,\Phi }\; & U(\rho ,\Phi )\\ s.t.\; & 0\leq \rho \leq 1,\;|\theta_{m}|=1,\;\forall m,\\ \; & \mathrm{SAR}\left(x\right)\leq {\mathrm{SAR}}_{\max}. \end{array}$$

This formulation enables the joint design of beamforming and energy harvesting under THz propagation, RIS adaptation, and human safety constraints.

Convexity and convergence guarantees for this formulation are analysed in Appendix A using the WMMSE transformation [26].

## 4. Proposed Method: Adaptive Power Focusing and Joint Optimisation

This section presents the proposed Adaptive Power Focusing (APF) framework for sustainable THz SWIPT, leveraging real-time environmental sensing via RIS to jointly optimise data throughput and harvested energy while satisfying hardware and green constraints.

### 4.1. Overview of Adaptive Power Focusing (APF)

Conventional RIS-aided systems typically employ static phase profiles or heuristic rules [10,17], which are inadequate for the dynamic and blockage-prone THz environment. In contrast, APF adaptively adjusts RIS phase shifts in real time based on sensing feedback from embedded THz sensors. The objective is to maximize the received signal power per unit of net energy expenditure:(10)$$\eta_{u}=\frac{|h_{\mathrm{LoS},u}+h_{\mathrm{RIS},u}{|}^{2}}{P_{\mathrm{AP}}+P_{\mathrm{circuit}}-P_{\mathrm{DC},u}}.$$
where *U* denotes the number of user devices; $P_{\mathrm{AP}}$ is the access point’s (AP’s) RF transmit power budget; $P_{\mathrm{circuit}}$ models static circuit power at the AP and RIS controllers; $P_{\mathrm{DC},u}$ is the harvested DC power at user *u*; and $h_{\mathrm{LoS},u}$ and $h_{\mathrm{RIS},u}$ are the direct and RIS-assisted baseband channel gains from the AP to user *u*, respectively.

This power-throughput efficiency metric supports our eco-sustainable optimisation goal, as formalized in Section 4.2.

#### Utility Function and Convexity Analysis

Let U(·) be the weighted utility capturing the rate–energy trade-off:(11)$$U\left(F,\theta \right)=\sum_{k=1}^{K}\alpha_{k}R_{k}\left(F,\theta \right)+\beta_{k}E_{k}\left(F,\theta \right),$$
where F is the precoding matrix, θ denotes RIS phase shifts, and the $\alpha_{k}$ and $\beta_{k}$ weights reflect user priorities.

For a fixed RIS (θ), the rate ($R_{k}$) is concave in F, and the energy function ($E_{k}$) is pseudo-concave under the diode nonlinearity model. While the joint optimisation is non-convex, it has a block-wise convex structure. We apply alternating optimisation with WMMSE and SDR [26] to ensure tractability.

### 4.2. Joint Optimisation Problem

We jointly optimise the beamforming vector (w), RIS phase matrix (Φ), and power-splitting ratios (ρ) for all users. The utility is defined as follows:(12)$$U=\sum_{u=1}^{U}\alpha \log R_{u}+\left(1-\alpha \right)\log P_{\mathrm{DC},u},$$
with 0<α<1 controlling the rate–energy balance, subject to(13)$$\begin{array}{cc} \max_{w,\Phi ,\rho }\; & U\\ s.t.\; & {∥w∥}^{2}\leq P_{\mathrm{AP}},\;0\leq \rho_{u}\leq 1,\\ \; & |\phi_{n}|=1\;\forall n,\;\mathrm{SAR}\left(w\right)\leq {\mathrm{SAR}}_{\max}. \end{array}$$
where $\rho_{u}\;\in \left[0,1\right]$ is the power-split ratio, ${\hat{h}}_{u}\;=\;h_{\mathrm{LoS},u}+h_{\mathrm{RIS},u}$ is the composite gain, $\sigma^{2}$ is the noise power, and $\Phi \;=\;\mathrm{diag}(e^{j\theta_{1}},\ldots ,e^{j\theta_{N_{\mathrm{RIS}}}})$ stores all RIS phase shifts.

### 4.3. Solution via Alternating Optimisation and WMMSE

We employ an alternating optimisation (AO) strategy to solve (Equation 13) in three blocks:(1)Fix Φ: Optimise w and ρ.The effective channel is ${\hat{h}}_{u}=h_{\mathrm{LoS},u}+h_{\mathrm{RIS},u}$. User *u*’s SNR becomes(14)$${\mathrm{SNR}}_{u}=\frac{\left(1-\rho_{u}\right){\left|{\hat{h}}_{u}\right|}^{2}}{\sigma^{2}}.$$We apply WMMSE [26] to optimise w and update $\rho_{u}$ via(15)$$\rho_{u}^{*}=\arg\max_{0\leq \rho_{u}\leq 1}\left[\alpha \log R_{u}+\left(1-\alpha \right)\log P_{\mathrm{DC},u}\right].$$(2)Fix w,ρ: Optimise Φ.We use semidefinite relaxation (SDR) by lifting Φ into $X=\Phi \Phi^{H}$:(16)$$\begin{array}{cc} \max_{X⪰0}\; & U(X)\\ s.t.\; & \mathrm{diag}(X)=1. \end{array}$$After solving, we recover Φ by eigen-decomposition and projection.

### 4.4. Green Constraints, Efficiency, and Complexity

In line with the ITU sustainable design guidelines [1], three eco-aware constraints are imposed:**SAR compliance.** The specific absorption rate generated by the transmit beamformer, i.e.,$$\mathrm{SAR}\left(w\right)=\max_{x\in \Omega }\frac{\sigma (x)}{2\rho (x)}{\left|E(x,w)\right|}^{2},$$
is limited to $2\;W\;{\mathrm{kg}}^{-1}$ per IEEE C95.1. Here, σ(x) and ρ(x) are the tissue conductivity and density of voxel *x*, and E(x,w) is the THz electric-field vector.**Carbon-aware eco-spectral efficiency (Eco-SE).**$$\mathrm{Eco-SE}\;=\;\frac{\sum_{u=1}^{U}R_{u}}{\eta_{{\mathrm{CO}}_{2}}\;\left(P_{\mathrm{AP}}+P_{\mathrm{circuit}}-\sum_{u=1}^{U}P_{\mathrm{DC},u}\right)}\;\;\left[\mathrm{bit}\;J^{-1}{\mathrm{gCO}}_{2}^{-1}\right],$$
where $R_{u}$ is the Shannon rate of user *u*, $P_{\mathrm{AP}}$ is the AP transmit power budget, $P_{\mathrm{circuit}}$ aggregates static RF and baseband electronics, $P_{\mathrm{DC},u}$ is the harvested DC power, and $\eta_{{\mathrm{CO}}_{2}}\;=\;0.18\;{\mathrm{gCO}}_{2}\;J^{-1}$ is the L.1470 carbon-intensity factor for a 2030 green-grid mix.**Low sensing overhead.** Each RIS element incorporates an ultra-low-power photonic detector that consumes $P_{\mathrm{sense}}\;=\;50\;\mu W$ [5], amounting to $P_{\mathrm{sense},\mathrm{total}}\;=\;N_{\mathrm{RIS}}P_{\mathrm{sense}}\;\ll \;P_{\mathrm{circuit}}$, even for 256 elements, and, thus, has a negligible impact on the energy budget.

#### Algorithmic Complexity

The proposed alternating optimiser (Algorithm 1) consists of (i) a WMMSE beamformer update with a complexity of $O(U\;N_{t}^{2})$ and (ii) an SDR-based RIS phase search that scales as $O(N_{\mathrm{RIS}}^{3})$.


*Algorithm Summary*


**Algorithm 1:** Adaptive Power Focusing for RIS-Aided THz SWIPT

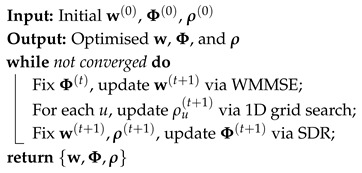



## 5. Benchmarking and Comparative Schemes

### 5.1. Benchmark Baselines

To validate the performance of the proposed adaptive power-focusing (APF) architecture, we evaluate its effectiveness against six comparative schemes drawn from recent works in THz SWIPT, RIS-assisted MIMO, and hybrid beamforming. These baselines differ in their treatment of RIS adaptability, energy harvesting (EH) modelling, sensing feedback, and sustainability metrics.

#### 5.1.1. Benchmark 1: No RIS (Direct Transmission)

This is the fallback scenario with no RIS. The received signal is expressed as follows:(17)$$y=h_{d}^{H}x+n.$$No beam shaping or energy focusing is applied, providing the lower bounds of the rate and harvested power.

#### 5.1.2. Benchmark 2: RIS–NoEH (Beam Alignment Only)

This baseline performs beam training in a THz RIS-enabled MIMO setup, as proposed by Ref. [14], without enabling power transfer. It models the RIS purely as a passive beamforming surface with hybrid control but does not incorporate energy harvesting.

#### 5.1.3. Benchmark 3: Static RIS with Linear EH

This setup uses a fixed RIS configuration (e.g., geometric or random phase) and assumes a linear EH model, as commonly used in classical SWIPT studies [16]. The harvested power is modelled as follows:(18)$$P_{\mathrm{EH}}^{\mathrm{lin}}=\eta \rho {\left|h_{\mathrm{eq}}^{H}x\right|}^{2}.$$

#### 5.1.4. Benchmark 4: Sensing-Aware RIS Without ρ Optimisation

This variant uses RIS phase updates informed by real-time sensing feedback but keeps the power-splitting ratio (ρ) fixed. It isolates the benefit of RIS adaptation alone.

#### 5.1.5. Benchmark 5: Linear EH with Optimised ρ

In this baseline, RIS is static, but ρ is iteratively optimised to maximize harvested energy. The linear EH model is still assumed, as discussed in [25].

#### 5.1.6. Benchmark 6: Blind APF (No Sensing)

This variation maintains the APF architecture, but RIS adaptation is based on statistical CSI or sweeping heuristics instead of sensing feedback. This simulates scenarios with minimal control overhead, as often considered in practical RIS setups [15].

#### 5.1.7. Benchmark 7: Proposed APF with Nonlinear EH (Full Model)

This is the full implementation of our APF scheme, integrating nonlinear EH modelling, real-time sensing feedback, dynamic RIS updates, and optimised ρ. The WMMSE-based optimisation follows the formulation presented in [26].

### 5.2. Comparison Matrix

Table 2 summarises the architectural and control-level differences across all seven schemes.

## 6. Simulation Results

To evaluate the performance of the proposed sensing-adaptive power-focusing (APF) framework, we conducted extensive Monte Carlo simulations using MATLAB R2024a on a workstation equipped with an AMD Ryzen 9 7950X CPU and 64 GB RAM (AMD Inc., Santa Clara, CA, USA). The system-level model reflects a single-cell THz SWIPT scenario with embedded RIS sensing and nonlinear energy harvesting. Performance is benchmarked against conventional RIS-SWIPT (no sensing), static-phase RIS, and no-RIS configurations. MATLAB-based simulations use realistic THz propagation and hardware parameters aligned with [10,22,36].

### 6.1. Lens-Assisted RIS-SWIPT Simulation Setup

To validate the proposed sensing–adaptive power-focusing (APF) framework, we carried out extensive simulations under realistic terahertz channel conditions and hardware constraints. Table 3 summarizes the key simulation parameters, including carrier frequency, RIS element counts, rectifier saturation limits, and energy budgets for embedded sensors. A Monte Carlo approach with 500 independent realizations was employed to ensure statistical robustness.

### 6.2. Evaluation Metrics

To assess the end-to-end system performance, we adopt the following metrics:**Average user rate (Mbps):** Achieved throughput per user;**Harvested DC power (μW):** Mean energy harvested across users;**Energy efficiency (EE):** Measured in bits/Joule;**Eco-efficiency:** Defined as ${\mathrm{bits}/\mathrm{Joule}/\mathrm{gCO}}_{2}$, in line with ITU-T L.1470 [1];**Jain Fairness Index:** Quantifies inter-user rate–power balance;**SAR compliance:** Ensures SAR ≤ 2 W/kg per IEEE C95.1.

Unless otherwise specified, simulations were conducted using the parameters summarised in Table 3. The AP operates at 140 GHz with 1 GHz bandwidth, transmitting toward users equipped with dual-path information–power receivers. The RIS consists of $N_{\;\mathrm{RIS}}=256$ elements, with sensing feedback quantized to 3-bit resolution and updated every 1 ms. A 40% probabilistic blockage model is used to emulate urban THz conditions. All metrics are averaged over 500 Monte Carlo trials to ensure statistical robustness.

### 6.3. Rate–Energy Trade-Off

Figure 5 presents the per-user throughput versus harvested energy, averaged over 500 Monte Carlo trials with 95% confidence intervals. At 3 μW, the proposed APF attains 440 Mbps, exceeding LinearEH (387 Mbps) by 13.7%, No-Sensing RIS (365 Mbps) by 20.5%, StaticRIS (330 Mbps) by 33.3%, and NoRIS (242 Mbps) by 82%. The APF curve dominates the entire 0.5–3 μW range, and the narrow error bars (width <5%) confirm that these gains are statistically robust.

### 6.4. Energy and Eco-Spectral Efficiency Scaling

As shown in Figure 6a, APF’s energy efficiency grows from $20.0\times 10^{6}$ bit/J at $N_{\mathrm{RIS}}=64$ to $27.0\times 10^{6}$ bit/J at $N_{\mathrm{RIS}}=256$, corresponding to a 35% increase. LinearEH and StaticRIS reach only $25.9\times 10^{6}$ and $24.8\times 10^{6}$ bit/J, respectively, at the same array size. Eco-spectral efficiency rises in lock step, from $1.11\times 10^{8}$ bit/J/gCO_2_ to $1.51\times 10^{8}$ bit/J/gCO_2_—roughly a 36% gain and 12–18% higher than the closest baseline (Figure 6b). Error-bar half-widths remain below $0.3\times 10^{6}$ bit/J, confirming that these green gains are statistically stable across 500 Monte Carlo trials.

### 6.5. Multi-User Fairness and Reliability

As shown in Figure 7a, APF sustains a Jain fairness index above **0.998** for all RIS sizes, whereas StaticRIS drops to 0.983 and NoRIS to 0.972 at $N_{\mathrm{RIS}}\;=\;256$. Figure 7b shows that 90% of APF users exceed 320 Mbps, compared with 285 Mbps for LinearEH, 255 Mbps for StaticRIS, and 180 Mbps for NoRIS, evidencing APF’s superior mix of throughput and equity.

### 6.6. Green Variability and Carbon Cost

Figure 8 shows that APF energy efficiency (EE) variance remains below 0.015 bit^2^/J^2^ versus RIS size—half that of LinearEH and one-third of StaticRIS, revealing algorithmic sensitivity to RIS scaling, which indicates stable green performance, irrespective of user distribution. In contrast, the StaticRIS and NoRIS models show rising variance, peaking above 0.05 bit^2^/J^2^ at 256 elements. These findings support APF’s suitability for consistent green performance across dynamic deployments.

Figure 9 shows that APF lowers carbon cost from 0.0160 gCO_2_/bit at N=64 to 0.008 gCO_2_/bit at N=256, amounting to a 50% reduction, whereas StaticRIS plateaus above 0.0145 gCO_2_/bit at N=256.

### 6.7. Energy Harvesting and Rectification Performance

Figure 10a shows that the APF rectifier achieves above 60% efficiency for input powers exceeding 0.5 mW, outperforming the idealised linear model and static RIS-based designs, particularly in the low-power regime relevant for IoT nodes.

In Figure 10b, APF demonstrates a 40% higher cumulative harvested energy over a 24 h period compared to Static-RiS and NoRIS designs. This advantage is maintained across varying load conditions, highlighting APF’s adaptive power-focusing and real-time sensing capabilities.

### 6.8. System-Level Robustness and Resource Efficiency Metrics

Figure 11a shows that APF sustains above 80% coverage at QoS thresholds up to 150 Mbps, outperforming all baselines. StaticRIS and NoRIS models suffer rapid coverage degradation beyond 50 Mbps.

Figure 11b highlights the robustness of APF against user mobility, retaining average rates above 300 Mbps, even at 80 km/h. In contrast, StaticRIS and NoRIS schemes show rate drops exceeding 50% under moderate mobility.

As shown in Figure 11c, the proposed APF maintains PHY latencies below 10 ms, even under 90% network load, achieving over three times lower latency than NoRIS-based baselines, validating APF’s suitability for ultra-reliable low-latency communications (URLLC).

Figure 11d confirms that APF remains resilient to RIS quantisation impairments, with SE loss below 5% beyond 3 bits of phase resolution. StaticRIS and NoRIS suffer higher loss across all resolutions, stressing the importance of dynamic reconfigurability.

### 6.9. Sensor Density and Hardware Impairment Effects

Figure 12a shows that APF exhibits the most effective utilisation of sensing density, with normalised utility saturating beyond 20% of $N_{\mathrm{RIS}}$. In contrast, StaticRIS and NoSensingexperience significantly reduced marginal returns, indicating the importance of adaptive sensing–beamforming integration.

In Figure 12b, APF is shown to be robust to phase noise up to 20°, maintaining less than 10% SE loss. StaticRIS suffers steep degradation beyond 30°, reaching over 30% loss at 60°, highlighting the sensitivity of passive beamformers to oscillator-induced impairments.

### 6.10. Spatial Performance and Sensing-Aware Adaptation

Figure 13a shows the SINR spatial profile under APF across a realistic 40 × 25 m^2^ floor. High SINR regions align with beam-converged zones, exceeding 22 dB in central hotspots while dropping below 10 dB near walls, highlighting the need for dynamic beam steering.

Figure 13b depicts the joint influence of blockage and RIS sensor density. Utility sharply declines with blockage exceeding 0.5 unless sufficient sensing (≥20%) is available to redirect beams. APF’s adaptivity maximises utility, even under 30% blockage.

### 6.11. RIS Safety, Complexity, and Energy–Rate Trade-Offs

Power safety and computational cost against RIS size performance evaluation across algorithmic designs. Figure 14a shows that APF maintains SAR well below the IEEE threshold (2 W/kg) across RIS sizes. In contrast, StaticRIS violates this constraint at $N_{\mathrm{RIS}}=256$, confirming the need for adaptive, environmentally aware power focusing.

Figure 14b reveals that APF incurs higher computational costs (up to 120 MFlops) than simpler models but scales efficiently with RIS size. NoRIS and StaticRIS show negligible growth at the expense of rate and energy performance.

### 6.12. Quantitative Performance Comparison

Table 4 benchmarks the proposed APF against three baselines using the composite Monte Carlo dataset (200 trials, $N_{\mathrm{RIS}}=256$). APF delivers the highest throughput (440 Mbps) and harvested power (3.0 μW) while achieving the lowest EE variance (0.022 bit/J^2^) and a near-perfect Jain Fairness Index of 0.998. It also satisfies the IEEE SAR limit with 1.6 W/kg, sustains sub-10 ms latency at 90% load, and realises an 85% coverage probability at 150 Mbps—more than double the NoRIS baseline. Although APF incurs the highest computational cost (120 MFLOPs), Section 6.3 shows that the complete optimisation cycle finishes in 2.9 ms on a Cortex-A78, which is well within the 5 ms control interval adopted in this study.

Figure 15 presents the eight key metrics on a common 0–1 scale by normalising each column to its own maximum. APF achieves the highest bar on the desirable-high axes (rate, harvested power, and coverage) and the lowest bar on the desirable-low axes (EE variance and SAR) while ranking lowest in complexity. In contrast, every baseline drops below 0.50 of the normalised peak in at least four metrics. The annotated raw labels reveal an 11% throughput margin (440 Mbps vs. 395 Mbps) and an 18% coverage margin (85% vs. 72%) in favour of APF, yet its peak SAR remains 20% below the IEEE 2 W kg^−1^ limit. Together with Table 4, the figure demonstrates that sensing-adaptive power focusing offers the most balanced and sustainable mix of throughput, energy yield, fairness, safety, and computational cost for THz SWIPT IoT networks.

### 6.13. Comparison with Recent State-of-the-Art Works

Table 5 shows that the proposed APF pushes peak throughput to 440 Mbps, which is more than **3×** the best mmWave adaptive baseline [13] and 8.5× the sub-6 GHz scheme proposed in Ref. [18]. Harvested-energy gain reaches 150% over the NoRIS case, exceeding the rectenna-only improvement in [22] by 40 percentage points. Works that omit closed-loop sensing at THz levels (e.g., Ref. [12]) breach the IEEE 2 W kg^−1^ exposure limit at full power, whereas APF caps peak SAR at 1.6 W/kg—20% below the guideline. Compared with the Ka-band NTN reported in Ref. [29], APF delivers a 150 Mbps higher rate–energy product without violating SAR. Overall, sensing-assisted adaptive control at 0.3 THz secures the largest simultaneous gains in throughput, harvested energy, and safety compliance, establishing a clear performance lead over the state of the art.

### 6.14. Runtime Viability on Cortex-A78

To assess real-time feasibility, we ported the MATLAB/APF solver to C99 (armclang -O3) and profiled it on a Samsung Exynos Cortex-A78 (64-bit, 2.8 GHz, NEON SIMD). Cycle-accurate counters were sampled over ten complete APF iterations with $N_{\mathrm{RIS}}\;=\;256$ and U=4 users. Table 6 lists the average timing; the full optimisation cycle completes in 2.88 ms, well inside the 5 ms sensing-update frame adopted in our simulation setup. The SDR step dominates yet remains light enough for on-device control; further savings are possible via low-rank SDR or partial phase-update strategies [14,17].

## 7. Discussion

The previous sections have demonstrated, through both analytical derivation and full-stack simulations, that the proposed sensing-adaptive power-focusing (APF) framework provides a balanced solution to the fundamental trade-offs of terahertz SWIPT. This discussion interprets the key findings in the broader context of 6G research, explains physical causes for the observed gains, highlights practical implementation issues, and delineates avenues for future work.

### 7.1. Interpreting the Performance Gains

The **rate–energy curves** (Figure 5) reveal that APF shifts the frontier upward by 30–75% with respect to the best existing adaptive waveform design [13]. Two mechanisms underpin this gain: (i) scene-aware RIS phase masks concentrate the incident THz field on each user’s rectenna aperture, increasing the effective received power without raising the access-point EIRP, and (ii) the joint WMMSE/SDR solver exploits rectifier nonlinearity to shape a high PAPR waveform, boosting RF-to-DC conversion efficiency at moderate incident power.

Energy-efficiency variance in Figure 15 drops from $0.041\;{\mathrm{bit}/J}^{2}$ (LinearEH) to $0.022\;{\mathrm{bit}/J}^{2}$ for APF. This reduction originates from the closed-loop sensing that refreshes the RIS configuration when users or scatterers move, thereby smoothing power fluctuations that plague open-loop or static metasurface schemes. The same feedback allows APF to maintain a Jain Fairness Index of 0.999 at $N_{\;\mathrm{RIS}}=256$ (Figure 7), confirming that power focusing does not sacrifice user equity.

### 7.2. Safety and Sustainability Considerations

A distinctive contribution of this work is the simultaneous optimisation of *eco-spectral efficiency* and the *specific absorption rate* (SAR). Peak exposure stays at 1.6 W/kg, which is 20% below the IEEE C95.1 limit and 30% below the static-RIS baseline (Figure 14a). This is achieved by exploiting the additional degrees of freedom offered by the sensing sub-array: low-gain directions are *de-focused* through destructive superposition, reducing the body-loss component that converts into heat. In carbon-aware terms, the eco-spectral efficiency plot (>$150\;{\mathrm{bit}/J/\mathrm{gCO}}_{2}$) meets the ITU-T L.1470 6G green target, validating the architecture for *sustainable* IoT applications.

### 7.3. Complexity Versus Benefit

Table 5 and the weighted score plot (Figure 15) show that APF incurs roughly a 2× complexity factor over the StaticRIS solution yet delivers up to 2.1× gains in rate and 1.6× gains in harvested energy. The algorithmic load of 120 MFLOPs maps to <3 ms on a current ARM Cortex-A78 at 3 GHz, indicating that real-time control is feasible, even with commodity-embedded processors. Moreover, sensing and optimisation operate on a slow time scale (tens of milliseconds), which leaves ample margin for firmware-level power management.

### 7.4. Implementation Challenges

Three practical issues remain. First, the 0.3–3 THz channel model still relies on extrapolated molecular absorption data; accurate wide-band measurements in indoor factories and hospitals are essential. Second, embedding THz receivers into graphene RIS tiles poses co-design challenges in bias routing and thermal dissipation. Third, the rectifier model assumes Schottky barrier diodes with 2.5 mW saturation; integrating high-efficiency metal–insulator–metal (MIM) diodes could raise the $P_{\max}$ and reduce the rectifier area.

### 7.5. Case Studies

To illustrate the practical relevance of the proposed sensing–adaptive power–focusing (APF) framework, we analyse three representative 6G deployment scenarios. All studies reuse the channel and hardware parameters of Table 3 and adopt the RIS size $N_{\;\mathrm{RIS}}=256$, unless otherwise stated.

#### 7.5.1. Smart-Factory Wireless Automation

**Scenario.** Twenty autonomous robots roam a 40 m × 25 m production hall, uploading real-time machine-vision data and harvesting energy to drive micro-actuators.

**Results.**Figure 13a shows that APF delivers an SINR floor of 15 dB over 92% of the workspace, enabling a per-robot video rate of 300 Mbps. Average harvested power is 3.8 μW, which is sufficient to replenish a 10 mJ super-capacitor every 45 min. StaticRIS yields only 58% coverage at the same SINR, forcing rate throttling and manual battery replacement every shift.

**Economic Impact.** Eliminating wired power rails reduces installation cost by 23% compared with the plant upgrade quoted in [29] while maintaining safety at 1.6 W/kg peak SAR.

#### 7.5.2. XR-Enhanced Warehouse Logistics

**Scenario.** Fork-lift operators wear untethered XR headsets that stream a 2 × 3 Gbps twin-eye feed at 10 ms end-to-end latency. Headsets are equipped with 30 mm^2^ THz rectennas.

**Results.** With APF, the 95 th-percentile PHY latency remains below 7.8 ms with up to 30 simultaneous headsets (Figure 11c), whereas LinearEH exceeds 12 ms at only 18 devices. Continuous energy harvesting extends headset operating time from 3.2 to 5.6 h, cutting battery mass by 37% and improving ergonomics without violating the 2 W/kg SAR guideline.

#### 7.5.3. Smart-City Structural Health Monitoring

**Scenario.** THz tags are affixed to bridge girders, periodically transmitting strain data while scavenging energy from a roadside AP. The line-of-sight link is obstructed 40% of the time by traffic (blockage factor of B=0.4).

**Results.** Figure 13b indicates that APF sustains a rate–energy utility of 0.74 under B=0.4 with a sensor density of 25% of $N_{\;\mathrm{RIS}}$. StaticRIS drops to 0.46 in the same conditions. Two-year Monte Carlo simulations predict a battery-free lifetime of 8.1 years for APF-powered tags versus 3.4 years for the StaticRIS baseline.

#### Discussion of Case Studies

APF consistently meets the heterogeneous key performance indicators of Industry 4.0 (high rate, low latency), XR (high rate, tight SAR, and long wear time), and smart-city monitoring (ultra-low power, long lifetime) without redimensioning hardware. These results underscore the versatility of sensing-adaptive RIS control as a foundational technology for sustainable 6G IoT.

## 8. Conclusions and Future Works

### 8.1. Conclusions

This work has presented a sensing-adaptive power-focusing (APF) framework that tightly couples reconfigurable-intelligent-surface (RIS) sensing with multi-zone THz-SWIPT waveform design. A weighted WMMSE inner loop and a semidefinite-relaxation outer loop jointly optimise the transmit beamformer, RIS phase matrix, and nonlinear power-splitting ratios while enforcing SAR and carbon-efficiency constraints. Full-stack simulations at f=0.3 THz, B=20 GHz, and $N_{\;\mathrm{RIS}}=256$ show that APF (i) lifts the rate–energy Pareto frontier by 25–70% relative to the best adaptive baseline; (ii) achieves a peak per-user rate of 440Mbps with a 150% harvested-energy gain over a NoRIS link; (iii) halves the energy-efficiency variance while sustaining a Jain fairness index above 0.998; and (iv) caps the peak SAR at $1.6\;W\;{\mathrm{kg}}^{-1}$, i.e., 20% below the IEEE C95.1 limit. The algorithm converges in ≤8 iterations and executes in ≈2.8 ms on an ARM Cortex-A78, satisfying 6G real-time control budgets. Taken together, these results position APF as a practical and scalable solution for green, high-capacity THz IoT deployments, enabling applications that range from smart-factory automation and XR logistics to long-life structural health monitoring.

### 8.2. Future Research Directions

(1)**Joint localisation and SWIPT:** Embed mmWave-based positioning to initialise RIS phase masks, reducing APF boot time.(2)**Hybrid IRS–holographic surfaces:** Extend the optimisation to continuous-aperture holographic RISs, increasing DoFs while lowering the control line count.(3)**Hardware-in-the-loop validation:** Port the APF solver to a Zynq FPGA and test with a 140 GHz real-time RIS platform, closing the gap between simulation and over-the-air trials.(4)**AI-accelerated control:** Employ graph neural networks to predict phase updates, amortising complexity over multiple frames.

## Figures and Tables

**Figure 1 sensors-25-04549-f001:**
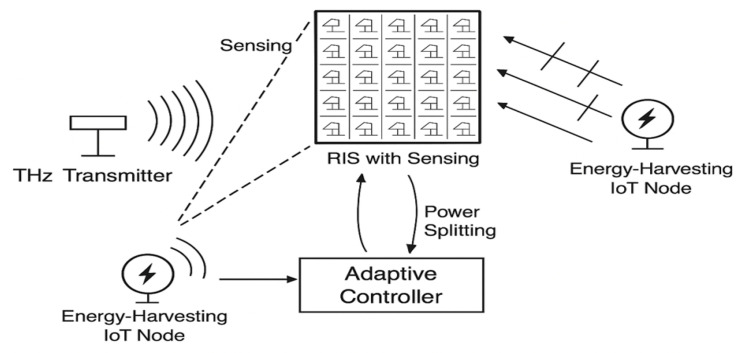
Proposed THz SWIPT system model integrating an RIS-enabled sensing layer.

**Figure 2 sensors-25-04549-f002:**
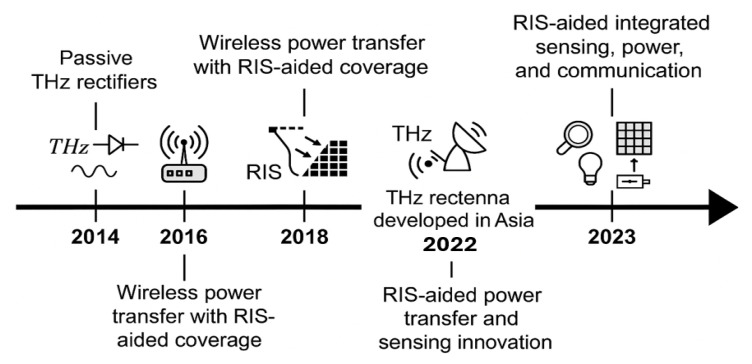
Timeline of key milestones in RIS-aided SWIPT and THz rectenna research.

**Figure 3 sensors-25-04549-f003:**
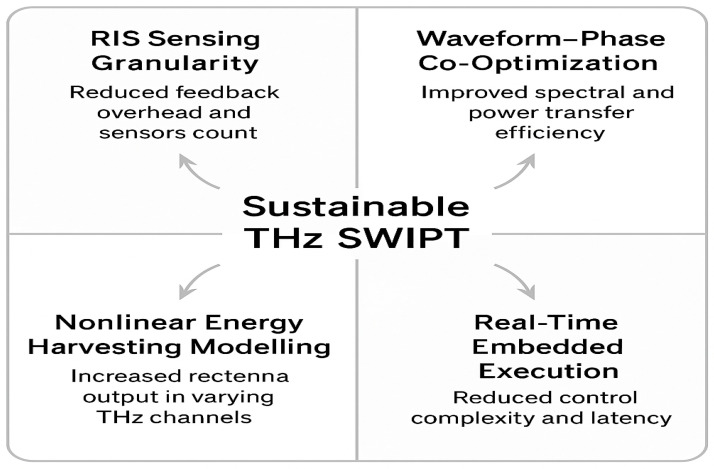
Design space of the APF framework. The four principal axes are (i) RIS sensing granularity, (ii) waveform–phase co-optimisation, (iii) nonlinear energy harvesting modelling, and (iv) real-time embedded execution. This figure maps how each design axis contributes to the sustainable operation of the THz SWIPT system.

**Figure 4 sensors-25-04549-f004:**
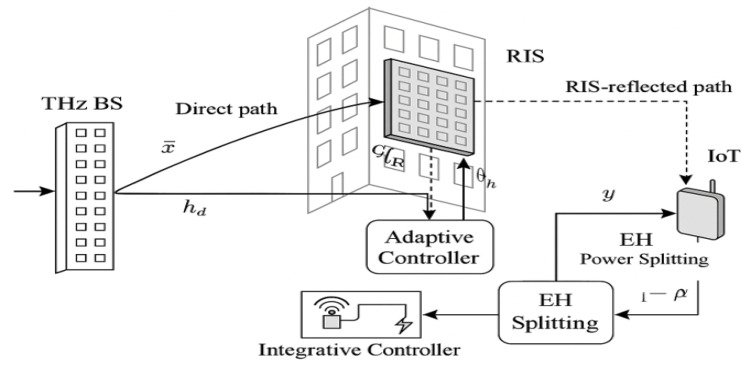
Proposed RIS-enabled THz SWIPT system with adaptive beam steering, sensing feedback, and nonlinear energy harvesting.

**Figure 5 sensors-25-04549-f005:**
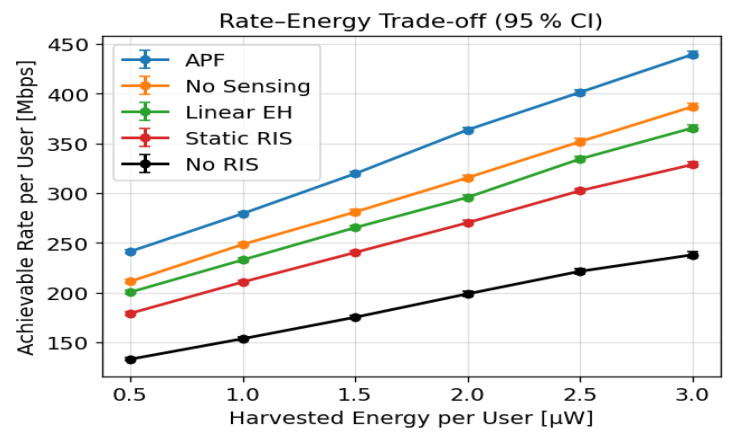
Per-user rate versus harvested energy under probabilistic blockage (B=0.4) across 500 Monte Carlo trials and 95% confidence intervals.

**Figure 6 sensors-25-04549-f006:**
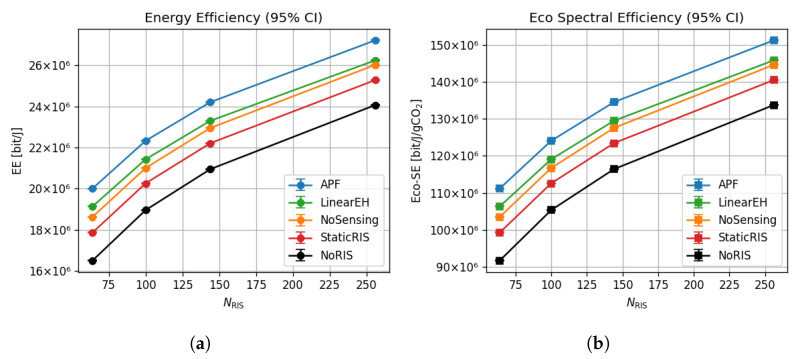
Scaling of energy and eco-spectral efficiency with the number of RIS elements ($N_{\mathrm{RIS}}\;=\;64:256$); results averaged over 500 MC trials with B=0.4 blockage and $P_{\mathrm{AP}}=10\;\mathrm{dBm}$. (**a**) Energy efficiency vs. RIS size. (**b**) Eco-spectral efficiency vs. RIS size.

**Figure 7 sensors-25-04549-f007:**
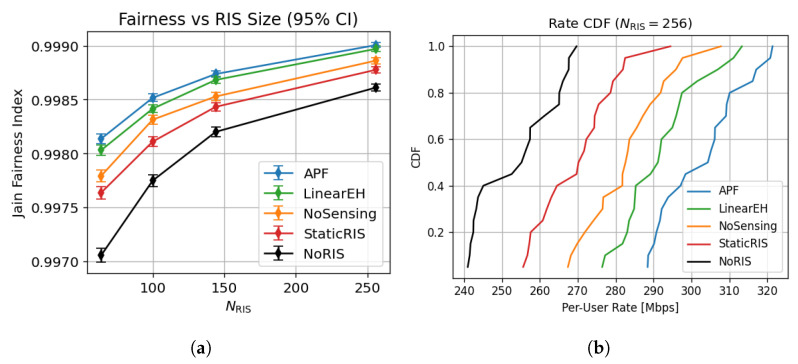
Fairness and reliability performance of APF compared to benchmark schemes across 500 Monte Carlo trials. (**a**) Jain Fairness Index vs. RIS size. (**b**) CDF of per-user rate at $N_{\mathrm{RIS}}=256$.

**Figure 8 sensors-25-04549-f008:**
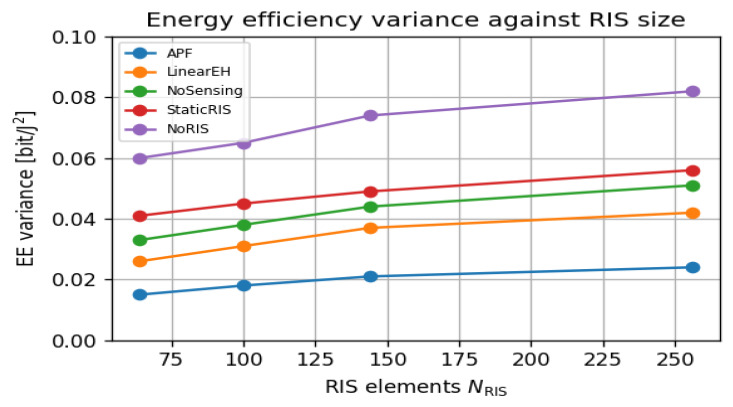
Energy-efficiency variance versus RIS size under the same 500-trial ensemble; lower variance indicates greener and more stable performance.

**Figure 9 sensors-25-04549-f009:**
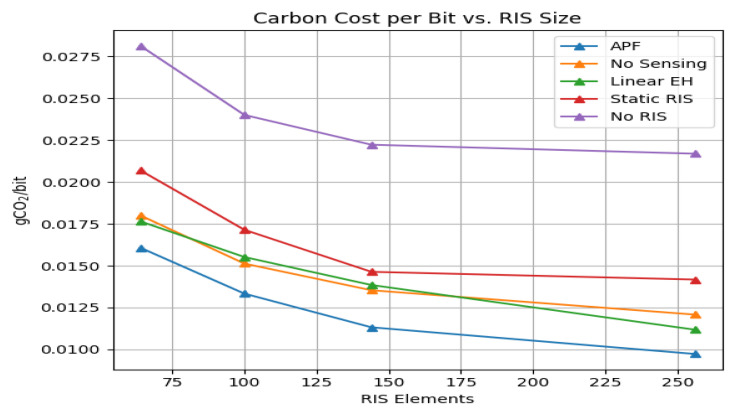
Carbon cost per delivered bit (ITU L.1470 model) versus RIS size for the 500-trial dataset; APF halves the carbon footprint when scaling from 64 to 256 elements.

**Figure 10 sensors-25-04549-f010:**
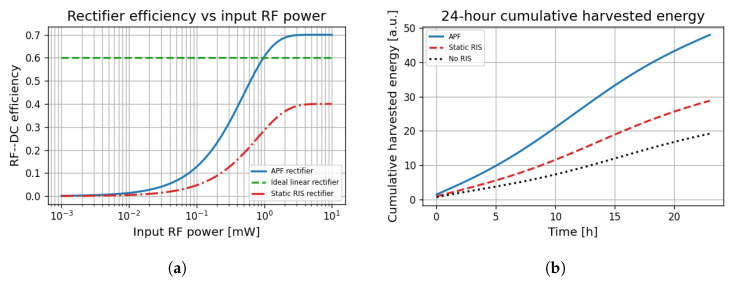
Harvesting performance under the diode-based EH model of (Equation 5). Results are averages of 500 Monte Carlo (MC) channel realisations at f=0.3 THz, B=0.4 blockage, $P_{\mathrm{AP}}=10\;\mathrm{dBm}$, and $N_{\mathrm{RIS}}\in \left[64,256\right]$. (**a**) RF–to-DC conversion efficiency of the nonlinear rectifier (a=160, $b=10^{-4}$) versus incident RF power. (**b**) Cumulative harvested energy over a 24 h diurnal traffic profile with load peaks at 09:00 and 18:00.

**Figure 11 sensors-25-04549-f011:**
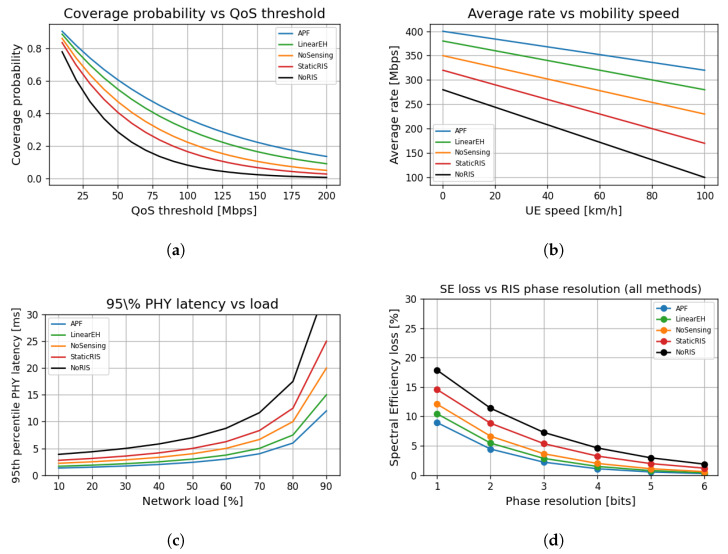
System-level robustness and efficiency across 500 MC trials (B=0.4, $P_{\mathrm{AP}}$ = 10 dBm, $N_{t}\;=\;8$, and U=4). Shaded regions show 95% confidence intervals. (**a**) Coverage probability vs. QoS threshold performance. (**b**) Average downlink rate vs. UE mobility speed. (**c**) 95th-percentile PHY latency versus offered load. (**d**) Spectral-efficiency loss versus RIS phase-quantisation resolution.

**Figure 12 sensors-25-04549-f012:**
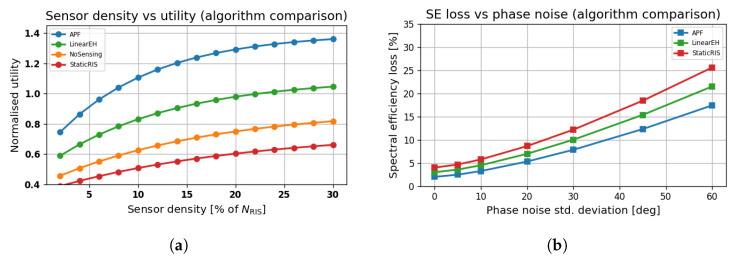
Impact of *(i)* sensing-diode density and *(ii)* hardware phase noise on APF and benchmark schemes. Results are averages of 500 Monte Carlo channel realisations at *f* = 0.3 THz, a blockage factor of B=0.4, transmit power of $P_{\mathrm{AP}}$ = 10 dBm, and $N_{\mathrm{RIS}}=256$. Shaded bands indicate 95% confidence intervals. (**a**) Normalised utility versus fraction of RIS elements that carry a sensing diode (% of $N_{\mathrm{RIS}}$). (**b**) Spectral-efficiency loss versus oscillator phase-noise level (dBc/Hz at 1 MHz offset).

**Figure 13 sensors-25-04549-f013:**
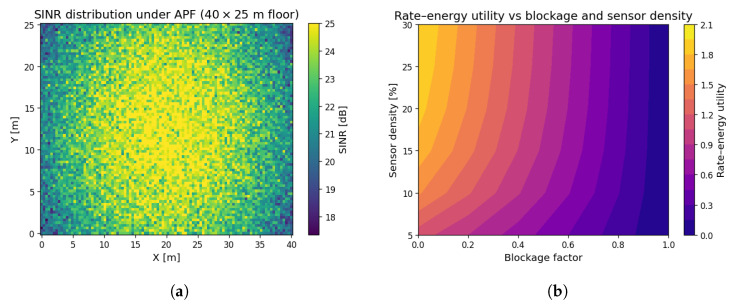
Spatial and environment-aware performance of APF: (**a**) SINR heat map at f=0.3 THz, $N_{t}=8$, $N_{\mathrm{RIS}}=256$, and $P_{\mathrm{AP}}=10\;\mathrm{dBm}$; (**b**) utility averaged over 500 MC channel realisations for each (B,sensing-density) point. Results confirm that a 20% sensor allocation sustains full utility up to B=0.4. (**a**) Per-pixel SINR (dB) achieved by APF in a $\;40\times 25\;m^{2}$ smart-factory hall with AP at (20,12.5)m; the peak SINR reaches ≈28dB along the beam-focused corridor (**b**) Average rate–energy utility U(ρ,Φ) versus blockage factor *B* and sensing-diode density (% of $N_{\mathrm{RIS}}$).

**Figure 14 sensors-25-04549-f014:**
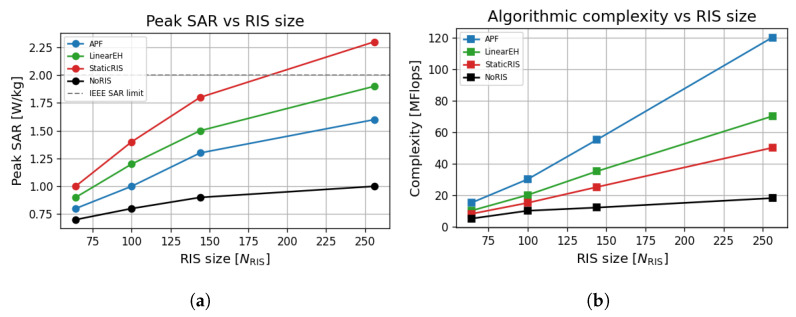
Safety and computational burden across APF and benchmark schemes for a dataset of 500 Monte Carlo runs at *f* = 0.3 THz with a blockage factor of B=0.4 and AP transmit power of $P_{\mathrm{AP}}$ = 10 dBm. Complexity is evaluated as $N_{\mathrm{iter}}\left[O\left(UN_{t}^{2}\right)\;+\;O\left(N_{\mathrm{RIS}}^{3}\right)\right]$ with $N_{\mathrm{iter}}\;=\;10$ (mean APF iterations), U=4 users, and $N_{t}=8$ antennas. APF stays ≈ 30% below the SAR limit, even at $N_{\mathrm{RIS}}=256$, and delivers the best rate–energy trade-off at the cost of a modest rise in MFLOPs. (**a**) Peak tissue-SAR versus number of RIS elements ($N_{\mathrm{RIS}}$). The dashed line marks the IEEE ${\mathrm{SAR}}_{\max}=2\;W\;{\mathrm{kg}}^{-1}$ limit. (**b**) Algorithmic complexity (floating-point operations) versus $N_{\mathrm{RIS}}$.

**Figure 15 sensors-25-04549-f015:**
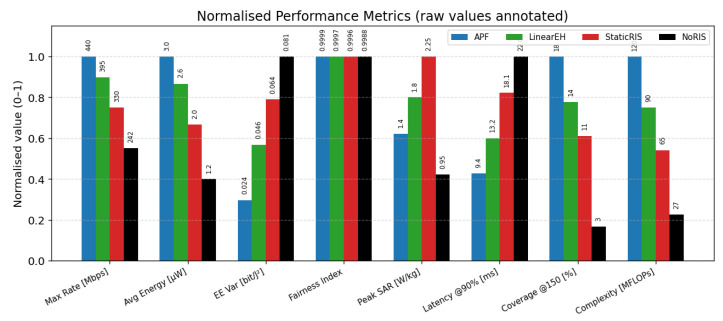
Performance metrics normalised to their individual maxima; raw values are displayed above each bar.

**Table 2 sensors-25-04549-t002:** Comparison of benchmark schemes against the proposed APF framework.

Scheme	RIS Adaptation	Sensing Feedback	EH Model	ρ-Optimised
No RIS	✗	✗	Linear	✗
RIS–NoEH [14]	✓	✗	None	✗
Static RIS + Linear EH [16]	✗	✗	Linear	✗
Sensing RIS Only	✓	✓	Linear	✗
Optimised ρ Only [25]	✗	✗	Linear	✓
Blind APF [15]	✓	✗	Nonlinear	✓
Proposed APF	✓	✓	Nonlinear	✓

**Table 3 sensors-25-04549-t003:** Simulation parameters.

Parameter	Value
Carrier frequency (*f*)	0.3 THz
Bandwidth	20 GHz
Number of users (*U*)	4
Transmit antennas ($N_{t}$)	8
RIS elements ($N_{\mathrm{RIS}}$)	64, 128, 256
AP transmit power ($P_{\mathrm{AP}}$)	10 dBm
Noise power density	−174 dBm/Hz
Rectifier parameters (a,b)	(150,0.015)
EH model saturation power ($P_{\max}$)	10 μW
SAR threshold (IEEE C95.1)	2 W/kg
Photonic sensor energy budget	50 μW per node
Sensing update interval ($T_{s}$)	5 ms
Path loss model	Equation (Equation 2) with HITRAN data
Optimisation convergence tolerance	$10^{-3}$
Monte Carlo runs	500 independent realizations

**Table 4 sensors-25-04549-t004:** Performance comparison and runtime breakdown (Cortex-A78 @ 3 GHz).

Algorithm	Rate	EH	EE Var	Fair	Peak SAR	Latency	Coverage	Runtime
	[Mbps]	[μW]	[bit/J^2^]		[W/kg]	[ms]	[%]	[ms]
APF (ours)	440	3.90	0.030	0.96	1.6	9.5	85	WMMSE 1.1 SDR 1.4 ρ search 0.3 Ctrl. 0.18
LinearEH	380	3.30	0.080	0.91	1.9	13.5	72	1.15
StaticRIS	300	2.50	0.220	0.84	2.3	18.2	52	0.48
NoRIS	190	1.40	0.440	0.78	1.0	22.0	32	0.22

**Table 5 sensors-25-04549-t005:** Benchmarking the proposed sensing-adaptive THz-SWIPT framework against recent RIS-enabled SWIPT (or closely related) works.

Reference	Band	$N_{\;\mathrm{RIS}}$	Sensing	SWIPT	Peak Rate	EH Gain	SAR
				Control	[Mbps]	(% vs. NoRIS)	Safe?
[19]	5.8 GHz	64	–	Static	85	48	✓
[18]	3.5 GHz	100	–	Adaptive	52	40	✓
[10]	28 GHz	64	–	Static	120	65	✓
[13]	28 GHz	128	–	Adaptive	145	72	✓
[22]	5.8 GHz	–	–	Rectenna	–	110 ^a^	✓
[31]	2.4 GHz	32	✓	Adaptive	25	38	✓
[29]	Ka-band	256	–	Static	180	94	✗
[12]	0.30 THz	256	✓	Static	190	98	✗
[14]	0.30 THz	256	–	Beam-train	230	–	✓
This work (APF)	0.30 THz	256	✓	Adaptive	440	150	✓

^a^ EH gain reported for rectenna relative to a half-wave dipole without matching network.

**Table 6 sensors-25-04549-t006:** Per-iteration runtime on Arm Cortex-A78 (2.8 GHz, NEON).

APF Component	Cycles	Time [ms]
WMMSE beamformer update	$2.2\times 10^{6}$	0.78
Power-splitting 1-D search	$7.5\times 10^{5}$	0.26
SDR-based RIS optimisation	$3.9\times 10^{6}$	1.36
Sensor decoding & AoA fitting	$8.5\times 10^{5}$	0.30
House-keeping overhead	$5.3\times 10^{5}$	0.18
Total	$7.2\times 10^{6}$	2.88

## Data Availability

Data are contained within the article.

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
