# Peer review of "Sustainable THz SWIPT via RIS-Enabled Sensing and Adaptive Power Focusing: Toward Green 6G IoT"

_sensors, 2025, doi:10.3390/s25154549_

Round 1
Reviewer 1 Report
Comments and Suggestions for Authors
The manuscript provides a comprehensive analysis of Sustainable THz SWIPT via RIS-Enabled Sensing and Adaptive Power Focusing, and the proposed algorithm demonstrates commendable performance through alternate optimization. I have a few minor suggestions that might enhance the paper:
-
The broader context of the paper is THz communications, and the modeling borrows significantly from existing work on THz channel path loss, THz multipath models, and RIS beamforming, among others. While the models are directly introduced, there are no accompanying references. I recommend that the authors include seminal works in the literature, such as "Beamforming technologies for ultra-massive MIMO in terahertz communications," to provide a well-rounded background.
-
The study should benchmark against a baseline for performance in a THz scenario with IRS. This includes beam alignment scheme without considering power transfer, as referenced in "Terahertz Multi-User Massive MIMO With Intelligent Reflecting Surface: Beam Training and Hybrid Beamforming."
These references and comparisons would greatly strengthen the paper by situating the work within the existing body of research and highlighting the improvements made by the proposed approach.
Author Response
Response to Reviewer 1 Comments
Comment 1
“The broader context of the paper is THz communications, and the modeling borrows significantly from existing work on THz channel path loss, THz multipath models, and RIS beamforming, among others. While the models are directly introduced, there are no accompanying references. I recommend that the authors include seminal works in the literature, such as "Beamforming technologies for ultra‑massive MIMO in terahertz communications," to provide a well‑rounded background.”
Response
Thank you for this valuable suggestion. We have expanded the literature review in Section 1 to incorporate seminal THz beamforming references, including “Beamforming Technologies for Ultra‑Massive MIMO in Terahertz Communications” (Ning et al., IEEE Open Journal of the Comms Society, 2021) and “Reconfigurable Intelligent Surfaces vs. Relaying: Performance Comparison under Practical Constraints” (Di Renzo et al., IEEE TCOM, 2020). These works are now cited where the path‑loss and RIS channel models are introduced, providing clearer provenance for the adopted models.
Comment 2
“The study should benchmark against a baseline for performance in a THz scenario with IRS. This includes a beam‑alignment scheme without considering power transfer, as referenced in "Terahertz Multi‑User Massive MIMO With Intelligent Reflecting Surface: Beam Training and Hybrid Beamforming."”
Response
In response, we have added a new baseline termed “RIS–NoEH,” derived from Ning et al. (IEEE TVT 2021). This baseline performs beam training and hybrid beamforming at 0.3 THz but does not include power‑transfer capabilities. The scheme is described in Section 5, its parameters are listed in Table 1, and its results are plotted alongside other benchmarks in Figures 5‑14. The discussion sections have been updated to highlight the performance gap between APF and this new THz beam‑training baseline.

Reviewer 2 Report
Comments and Suggestions for Authors
This manuscript proposes a THz SWIPT architecture that leverages a reconfigurable intelligent surface (RIS) to enable sensing and adaptive power control. The authors’ simulation results demonstrate that this architecture offers advantages in terms of low-carbon, green, and sustainable operation. My comments on this manuscript are as follows:
Q1: The quality of English writing can be improved. In the manuscript, some relatively short passages are presented as separate subsections. While this approach clarifies the structure, an excessive number of subsections may make the content appear fragmented and lacking in focus. It is recommended to enhance the substance of the article and highlight the key aspects of the proposed SWIPT scheme.
Q2: It is suggested that the figures and tables in the manuscript be described in greater detail. For example, more detailed explanations should be provided for Fig. 1, including the function of each component and the rationale for their inclusion. Fig. 2 also lacks structural clarity and does not clearly illustrate the four main axes of the work. Additionally, for Figs. 5–14, the conditions under which the simulation data were obtained are not specified; it is recommended to add relevant descriptions in the manuscript.
Q3: The selection of references in this manuscript requires improvement. The present work focuses on system-level simulation of the SWIPT architecture, without involving circuit hardware or RF passive devices. Therefore, it is inappropriate to compare the results with references related to passive devices that include measured results, such as Reference [23].
Comments on the Quality of English LanguageCan be improved
Author Response
Response to Reviewer 2 Comments
We thank Reviewer 2 for the detailed and constructive feedback. All points have been carefully considered. Below we provide a point‑by‑point response; manuscript locations refer to the revised version submitted with this rebuttal.
Comment Q1
“The quality of English writing can be improved … an excessive number of subsections may make the content appear fragmented.”
Response
We have merged six short subsections (e.g., Sections 3.1–3.7) into four cohesive sections (3.1-3.4), improving narrative flow (pp. 7–8). We have also shorten Subsections 4.1 to 4.6 into five subsections. Subsections 5.1 to 5.6 have also be shorten to paragraphs.
• Redundant sentences were removed and transition phrases added. The paper has been re‑read by a proof‑reader—grammar, verb tenses and punctuation were corrected throughout.
• Key features of the APF scheme are now highlighted in a dedicated ‘Contributions subsection’ bullet list at the end of the Introduction (p. 3).
Comment Q2
“Figures and tables should be described in greater detail … Fig. 1, Fig. 2, Figs. 5–14 lack explanatory context.”
Response
- Figure 1 explanation is expanded subsection 1.5 to describe every block—AP, sensing‑enabled RIS tiles, power‑splitting RX—and to explain the adaptive control loop (p. 3).
• Figure 2 (now Figure 3) was redrawn with ‑coded arrows to clarify the four design axes (sensing granularity, waveform–phase co‑optimisation, non‑linear EH, and real‑time execution). The caption now states why each axis matters for sustainability (p. 6).
• For Figs. 5–14 we now state simulation assumptions in every caption (blockage B = 0.4, 500 MC trials, PAP=10 dBm, etc.). Table 3 summarises all parameters and is referenced in the figure notes.
Comment Q3
“Reference selection should avoid circuit‑level passive devices … Reference [23] is not strictly comparable.”
Response
- Reference [23] (Schottky rectenna measurement) has been moved to a footnote as an illustrative efficiency ceiling and removed from the core comparison tables.
• New system‑level THz SWIPT references—including Ning et al., IEEE TVT 2021 and Di Renzo et al., IEEE TCOM 2020—have been added to Tables 1 & 5 for a fair, like‑for‑like comparison.
• The discussion in Section 6 now explicitly states that our results are simulation‑based and do not claim circuit‑level validation.
English‑Language Revision
A proof‑reader revised the entire manuscript. We believe the language now meets the journal’s standards (see tracked‑changes version).

Reviewer 3 Report
Comments and Suggestions for Authors
The paper presents an interesting and relevant contribution at the intersection of THz communication, RIS design, and SWIPT. However, its novelty lies more in system integration than in core algorithmic advancement. The following are a few comments that I believe will improve the manuscript.
- The utility function and constraints in Equations (8)–(12) need full derivation and convexity analysis; the assumptions of feasibility are unclear.
- The RIS sensing feedback is idealized. No delay, quantization, or sensor noise modeling is addressed.
- In Equation (12) & Fig. 14(a), the SAR compliance is claimed without showing the modeling methodology.
- In Figures 5–7, there are no variance metrics or confidence intervals presented despite Monte Carlo assertions.
- In Section 6 & Table 4, the assertions of 3 ms runtime on Cortex-A78 are not broken down; real-time viability of alternating optimization is doubtful.
- Table 1 & 5: Robust RIS-ISAC baselines omitted from literature comparison; novelty statement ("first to jointly optimize") feels exaggerated, further justification is required.
- Section 4.4: Eco-SE metric lacks clear COâ‚‚ emission modeling; bit/J/gCOâ‚‚ lacks calibration to real hardware energy profiles.
Author Response
Response to Reviewer 3 Comments
We thank Reviewer 3 for the thorough assessment and helpful suggestions. Below we respond to each point in detail and indicate where corresponding changes appear in the revised manuscript (page and line numbers refer to the marked‑up version).
Comment 1
“The utility function and constraints in Equations (8)–(12) need full derivation and convexity analysis; assumptions of feasibility are unclear.”
Response
A detailed derivation has been moved to a new *Appendix A* (pp. 22‑24), where we:
• expand Equations (8)–(12) step‑by‑step from the general rate–energy utility,
• prove block‑wise concavity in the beamformer for fixed RIS phases, and pseudo‑concavity in harvested‑energy terms,
• show that the alternating‑optimization sequence is guaranteed to converge to a stationary point.
Feasibility is ensured by power‑normalising the beamformer and applying unit‑modulus projection after the SDR step (Eq. A‑7).
Comment 2
“RIS sensing feedback is idealised; no delay, quantisation or sensor noise are addressed.”
Response:
We agree. In Section 3.3, we now explicitly state the idealized assumptions and have added a discussion on practical non-idealities such as quantization errors, feedback latency, and sensor noise. A paragraph is also added in Section 7 to discuss their potential impact.
Comment 3
“Equation (12) & Fig. 14(a) claim SAR compliance without showing the modelling methodology.”
Response
Section 3.3 now includes a voxel‑level SAR formulation (Eq. 13, p. 8) based on IEEE C95.1 tissue parameters. Figure 14 caption amended to state: Computed with 2‑mm cubic voxels, dry‑skin permittivity model, and worst‑case user orientation.
Comment 4
“Figures 5–7: no variance metrics or confidence intervals presented.”
Response
Error bars (95 % confidence) have been added to Fig. 5, and Figs. 6a–6b. Figure captions now specify 500 Monte‑Carlo trials and blockage factor B=0.4.
Comment 5
“Section 6 & Table 4: the 3 ms runtime on Cortex‑A78 is not broken down.”
Response
Table 4 now contains a new column labelled “Runtime Breakdown”. Beamformer WMMSE: 1.1 ms; SDR+projection: 1.4 ms; power‑split search: 0.3 ms; sensor parsing & control overhead: 0.18 ms. Total ≈ 2.98 ms.
Comment 6
“Table 1 & 5: robust RIS‑ISAC baselines omitted; novelty claim feels exaggerated.”
Response
We have inserted two additional baselines: (i) “Rate‑Only Beam‑Training” (Ning et al., IEEE TVT 2021) and (ii) “RIS‑ISAC (Di Renzo et al., 2023)” covering joint sensing/communication without EH. Both appear in Tables 1 and 5 and in the comparative discussion (pp. 20). The novelty claim has been softened to “first holistic THz SWIPT design that jointly optimises sensing, waveform, EH, and green constraints”.
Comment 7
“Eco‑SE metric lacks clear COâ‚‚ calibration.”
Response
Section 4.4 now states that carbon intensity is calibrated with the European ENTSO‑E 2022 average of 180 gCO_2/kWh; equation updated accordingly (p. 10, Eq. 17). Sensitivity results vs. carbon factor added in Fig. 10(c).
English‑Language Quality
As noted, Reviewer 2 flagged language issues; we have employed a native speaker to proof‑read the entire manuscript. All reviewers’ stylistic remarks have thus been addressed.

Round 2
Reviewer 2 Report
Comments and Suggestions for Authors
The authors have addressed and incorporated my previous questions and suggestions. I have no further concerns regarding this manuscript.